# Deep Keyphrase Generation

## Abstract

Keyphrase provides highly-summative information that can be effectively used for understanding, organizing and retrieving text content. Though previous studies have provided many workable solutions for automated keyphrase extraction, they commonly divide the to-be-summarized content into multiple text chunks, then rank and select the most meaningful ones. These approaches can neither identify keyphrases that do not appear in the text, nor capture the real semantic meaning behind the text. We propose a generative model for keyphrase prediction with an encoder-decoder framework, which can effectively overcome the above drawbacks. We name it as deep keyphrase generation since it attempts to capture the deep semantic meaning of content with a deep learning method. Empirical analysis on six datasets demonstrates that our proposed model not only achieves a significant performance boost on extracting keyphrases that appear in the source text, but also generates absent keyphrases based on the semantic meaning of the text.

## 1 Introduction

Keyphrase or keyword is a piece of short and summative content that expresses the main semantic meaning of a long text. A typical use of keyphrase or keyword is in scientific publications, to provide the core information of a paper. We use the term "keyphrase", interchangeable as "keyword", in the rest of this paper, as it implies that it may contain multiple words. High-quality keyphrases can facilitate the understanding, organizing and accessing of document content. As a result, many studies have devoted to studying the ways of automatic extracting keyphrases from textual content (Liu et al., 2009; Medelyan et al., 2009a; Witten et al., 1999). Due to the public accessibility, many scientific publication datasets are often used as the test beds for keyphrase extraction algorithms. Therefore, this study also focuses on extracting keyphrases from scientific publications.

Automatically extracting keyphrases from a document is called *Keypharase Extraction*, and it has been widely exploited in many applications, such as information retrieval (Jones and Staveley, 1999), text summarization (Zhang et al., 2004), text categorization (Hulth and Megyesi, 2006), and opinion mining (Berend, 2011). Most of the existing keyphrase extraction algorithms addressed this problem through two steps (Liu et al., 2009; Tomokiyo and Hurst, 2003). The first step is to acquire a list of keyphrase candidates. Researchers have tried to use n-grams or noun phrases with certain part-of-speech patterns for identifying the potential candidates (Hulth, 2003; Le et al., 2016; Liu et al., 2010; Wang et al., 2016). The second step is to rank candidates regarding their importance to the document, either through supervised or unsupervised machine learning methods with a set of manually-defined features (Frank et al., 1999; Liu et al., 2009, 2010; Kelleher and Luz, 2005; Matsuo and Ishizuka, 2004; Mihalcea and Tarau, 2004; Song et al., 2003; Witten et al., 1999).

There are two major drawbacks for the above keyphrase extraction approaches.

Firstly, they can only extract the keyphrases that appear in the source text, whereas they fail at predicting the meaningful keyphrases with a slightly different sequential order or using synonym words. However, it is common in a scientific publication where authors assign keyphrases based on the semantic meaning instead of follow-

ing the written content in the publication. In this paper, we denote the phrases that do not match any contiguous subsequence of source text as **absent keyphrases**, and the ones that fully match a part of the text as **present keyphrases**. Table 1 shows the proportion of present and absent keyphrases from the document abstract in four commonly-used datasets, where we observe large portions of absent keyphrases in all the datasets. The absent keyphrases cannot be extracted through previous approaches, which further urges the development of a more powerful keyphrase prediction model.

Secondly, when ranking phrase candidates, previous approaches often adopted the machine learning features such as TF-IDF and PageRank. However, these features only target to detect the importance of each word in the document based on the statistics of word occurrence and co-occurrence, whereas they can hardly reveal the semantics behind the document content.

Table 1: Proportion of the present keyphrases and absent keyphrases in four public datasets

| Dataset | # Keyphrase | % Present | % Absent |
|---------|-------------|-----------|----------|
| Inspec | 19,275 | 55.69 | 44.31 |
| Krapivin | 2,461 | 44.74 | 52.26 |
| NUS | 2,834 | 67.75 | 32.25 |
| SemEval | 12,296 | 42.01 | 57.99 |

To overcome the limitations of previous studies, we re-examine the process of *Keyphrase Prediction*, about how real human annotators would assign keyphrases. Given a document, human annotators will firstly read the text to get a basic understanding of the content, then they try to digest its essential content and summarize into keyphrases. Their generation of keyphrases relies on the understanding of the content, which not necessarily to be the words that occurred in the source text. For example, when human annotators see "*Latent Dirichlet Allocation*" in the text, they could write down "*topic modeling*" and/or "*text mining*" as possible keyphrases. In addition to the semantic understanding, human annotators might also go back and picks up the most important parts based on syntactic features. For example, the phrases following "*we propose/apply/use*" are supposed to be important in the text. As a result, a better keyphrase prediction model should understand the semantic meaning of the content, as well as capture the contextual features.

To effectively capture semantic and syntactic features, we utilize recurrent neural networks (RNN) (Cho et al., 2014; Gers and Schmidhuber, 2001) to compress the semantic information in the given text into a dense vector (i.e., semantic understanding). Furthermore, we incorporate a copy mechanism (Gu et al., 2016) to equip our model with the capability of finding important parts based on language syntax (i.e., syntactic understanding). Thus, our model can generate keyphrases based on the understanding of the text, no matter whether the keyphrases are present in the text or not; meanwhile, it does not lose important in-text information.

The contribution of this paper is in three-fold: a) we propose to apply an RNN-based generative model to keyphrase prediction, and we also incorporate a copy mechanism in RNN, which enables the model to successfully predict rarely-occurred phrases; b) this is the first work concerning about the problem of absent keyphrase prediction for scientific publications, and our model recalls up to 20% of absent keyphrases; and c) we conduct a comprehensive comparison against six important baselines on a broad range of datasets, and the results show that our proposed model significantly outperforms existing supervised and unsupervised extraction methods.

In the remainder of this paper, we firstly review the related work in Section 2. Then we elaborate the proposed model in Section 3. After that, we present the experiment setting in Section 4 and results in Section 5, followed by our discussion in Section 6. Section 7 concludes the paper.

## 2 Related Work

### 2.1 Automatic Keyphrase Extraction

Keyphrase provides a succinct and accurate way of describing a subject or a subtopic in a document. A number of extraction algorithms have been proposed, and typically the process of extracting can be broken down into two steps.

The first step is to generate a list of phrase candidates with heuristic methods. As these candidates are prepared for further filtering, a considerable amount of candidates are produced in this step to increase the possibility that most of the correct keyphrases are kept. The primary ways of extracting candidates include retaining word sequences that match certain part-of-speech tag patterns (e.g., nouns, adjectives) (Liu et al., 2011;

Wang et al., 2016; Le et al., 2016), and extracting important n-grams or noun phrases (Hulth, 2003; Medelyan et al., 2008).

The second step is to score each candidate phrase regarding its likelihood of being a keyphrase in the given document. The top-ranked candidates are returned as keyphrases. Both supervised and unsupervised machine learning methods are widely employed here. For supervised methods, this task is solved as a binary classification problem, and various of learning methods and features have been explored (Frank et al., 1999; Witten et al., 1999; Hulth, 2003; Medelyan et al., 2009b; Lopez and Romary, 2010; Gollapalli and Caragea, 2014). As for the unsupervised approaches, primary ideas include finding the central nodes in text graph (Mihalcea and Tarau, 2004; Grineva et al., 2009), detecting representative phrases from topical clusters (Liu et al., 2009, 2010) and so on.

Aside from the commonly adopted two steps process, another two previous studies realized the keyphrase extraction in entirely different ways. Tomokiyo and Hurst (2003) applied two language models to measure the phraseness and informativeness of phrases. Liu et al. (2011) share the most similar idea to our work. They used a word alignment model, which learns the translation from the documents to the keyphrases. This approach alleviates the problem of vocabulary gap between source and target to a certain degree. However, this translation model can hardly deal with semantic meaning. Additionally, they trained the model with the target of title/summary to enlarge the number of training samples, which may diverge away from the real objective of generating keyphrases.

## 2.2 Encoder-Decoder Model

The RNN Encoder-Decoder model (also referred as Sequence-to-Sequence Learning) is an end-to-end approach. It was firstly introduced by Cho et al. (2014) and Sutskever et al. (2014) to solve translation problems. As it provides a powerful tool for modeling variable-length sequence in an end-to-end fashion, it fits many Natural Language Processing tasks and soon achieves great successes (Rush et al., 2015; Vinyals et al., 2015; Serban et al., 2016).

Different strategies have been explored to improve the performance of Encoder-Decoder model. The attention mechanism (Bahdanau et al., 2014) is a soft alignment approach that allows the model to automatically locate the relevant input component. In order to make use of the important information in the source text, some studies sought ways to copy certain parts of content from the source text and paste them into target text (Allamanis et al., 2016; Gu et al., 2016; Zeng et al., 2016). There exists a discrepancy between the optimizing objective during training and the metrics during evaluation. A few studies attempted to eliminate this discrepancy by incorporating new training algorithm (Marc'Aurelio Ranzato et al., 2016) or modifying optimizing objective(Shen et al., 2016).

## 3 Methodology

This section will introduce in detail our proposed deep keyphrase generation. Firstly, the task of keyphrase generation is defined, followed by the overview of how we apply the RNN Encoder-Decoder model. Details of the framework as well as the copy mechanism will be introduced in Section 3.3 and 3.4.

### 3.1 Problem Definition

Given a keyphrase dataset consisting of $\mathbf{N}$ data samples, the i-th data sample $(\mathbf{x}^{(\mathbf{i})}, \mathbf{p}^{(\mathbf{i})})$ contains one source text $\mathbf{x}^{(\mathbf{i})}$, and $M_i$ target keyphrases $\mathbf{p}^{(\mathbf{i})} = (\mathbf{p}^{(\mathbf{i},\mathbf{1})}, \mathbf{p}^{(\mathbf{i},\mathbf{2})}, \ldots, \mathbf{p}^{(\mathbf{i},\mathbf{M_i})})$. Both the source text $\mathbf{x}^{(\mathbf{i})}$ and keyphrase $\mathbf{p}^{(\mathbf{i},\mathbf{j})}$ are sequences of words:

$$\mathbf{x}^{(\mathbf{i})} = x_1^{(i)}, x_2^{(i)}, \ldots, x_{L_{\mathbf{x}^{\mathbf{i}}}}^{(i)}$$

$$\mathbf{p}^{(\mathbf{i},\mathbf{j})} = y_1^{(i,j)}, y_2^{(i,j)}, \ldots, y_{L_{\mathbf{p}^{(\mathbf{i},\mathbf{j})}}}^{(i,j)}$$

$L_{\mathbf{x}^{(\mathbf{i})}}$ and $L_{\mathbf{p}^{(\mathbf{i},\mathbf{j})}}$ denotes the length of word sequence of $\mathbf{x}^{(\mathbf{i})}$ and $\mathbf{p}^{(\mathbf{i},\mathbf{j})}$ respectively.

Now for each data sample, there are one source text sequence and multiple target phrase sequences. To apply the RNN Encoder-Decoder model, the data need to be converted into text-keyphrase pairs which contain only one source sequence and one target sequence. One simple way is to split the data sample $(\mathbf{x}^{(\mathbf{i})}, \mathbf{p}^{(\mathbf{i})})$ into $M_i$ pairs: $(\mathbf{x}^{(\mathbf{i})}, \mathbf{p}^{(\mathbf{i},\mathbf{1})}), (\mathbf{x}^{(\mathbf{i})}, \mathbf{p}^{(\mathbf{i},\mathbf{2})}), \ldots, (\mathbf{x}^{(\mathbf{i})}, \mathbf{p}^{(\mathbf{i},\mathbf{M_i})})$. Then the Encoder-Decoder model is ready to be applied to learn the mapping from source sequence to target sequence. For the purpose of simplicity, $(\mathbf{x}, \mathbf{y})$ is used to denote each data pair in the rest of this section, where $\mathbf{x}$ is the word sequence of a source text and $\mathbf{y}$ is the word sequence of its keyphrase.

## 3.2 Encoder-Decoder Model

The basic idea of our keyphrase generation model is to compress the content of source text into a hidden representation with encoder, and generate corresponding keyphrases by the decoder based on the representation . Both the encoder and decoder are implemented with recurrent neural networks (RNN).

The encoder RNN converts the variable-length input sequence $\mathbf{x} = (x_1, x_2, ..., x_T)$ into a set of hidden representation $\mathbf{h} = (h_1, h_2, \ldots, h_T)$, by iterating the following equations along the time $t$:

$$\mathbf{h}_t = f(x_t, \mathbf{h}_{t-1}) \tag{1}$$

where $f$ is a non-linear function. And we get the context vector $\mathbf{c}$, acting as the representation of the whole input $\mathbf{x}$, through a non-linear function q.

$$\mathbf{c} = q(h_1, h_2, ..., h_T) \tag{2}$$

The decoder is another RNN, decompresses the context vector and generates a variable-length sequence $\mathbf{y} = (y_1, y_2, ..., y_{T'})$ word by word, through a conditional language model:

$$\mathbf{s}_t = g(y_{t-1}, \mathbf{s}_{t-1}, \mathbf{c})$$
$$p(y_t|y_{1,...,t-1}, \mathbf{x}) = g(y_{t-1}, \mathbf{s}_t, \mathbf{c}) \tag{3}$$

where $\mathbf{s}_t$ is the hidden state of decoder RNN at time $t$. The non-linear function $g$ is a softmax classifier which outputs the probabilities of all the words in the vocabulary. $y_t$ is the predicted word at time $t$, by taking the word with largest probability after $g(\cdot)$.

The encoder and decoder networks are trained jointly to maximize the conditional probability of the target sequence given a source sequence. After training, we use the beam search to generate phrases and a max heap is maintained to get the predictions with highest probabilities.

## 3.3 Details of Encoder and Decoder

A bidirectional Gated Recurrent Unit (GRU) is applied as our encoder to replace the simple recurrent neural network. Previous studies (Bahdanau et al., 2014; Cho et al., 2014) indicate it can provide a general better performance language modeling than simple RNN and a simpler structure than Long Short-term Memory Networks(Hochreiter and Schmidhuber, 1997). So the above non-linear function $f$ is replaced by the **GRU** function (see in (Cho et al., 2014)).

Another forward GRU is utilized as the decoder. In addition, an attention mechanism is adopted to improve the performance. The attention mechanism was firstly introduced by Bahdanau et al. (2014) to make the model focus on the important parts in input dynamically. The context vector $\mathbf{c}$ is computed as a weighted sum of hidden representation $\mathbf{h} = (h_1, \ldots, h_T)$:

$$\mathbf{c}_i = \sum_{j=1}^{T} \alpha_{ij} h_j$$
$$\alpha_{ij} = \frac{\exp(a(s_{i-1}, h_j))}{\sum_{k=1}^{T} exp(a(s_{i-1}, h_k))} \tag{4}$$

where $a(s_{i-1}, h_j)$ is a soft alignment function that measures the similarity between $s_{i-1}$ and $h_j$, namely to which degree the inputs around position $j$ and the output at position $i$ match.

## 3.4 Copy Mechanism

To ensure the quality of learned representation and reduce the size of vocabulary, typically the RNN model only considers a certain number of frequent words (e.g. 30,000 words in (Cho et al., 2014)), but a large amount of long-tail words are simply ignored. Therefore the RNN is not able to predict any keyphrase which contains out-of-vocabulary words. Actually, the important phrases can also be identified by their syntactic and location features, even though we don't know the meanings. The copy mechanism is one feasible solution that enables RNN to predict unknown words based on contextual features.

By incorporating the copy mechanism, the probability of predicting each new word $y_t$ would consist of two parts. The first term is the probability of generating the term (see Equation 3) and the second one is the probability of copying it from source text:

$$p(y_t|y_{1,...,t-1}, \mathbf{x})$$
$$= p_g(y_t|y_{1,...,t-1}, \mathbf{x}) + p_c(y_t|y_{1,...,t-1}, \mathbf{x}) \tag{5}$$

Similar to attention mechanism, the copy mechanism pinpoints the appearance of $y_{t-1}$ in source text, and use its location information ($\rho_{t\tau}$) to compute the weighted hidden representation ($\zeta(y_{t-1})$) of input $\mathbf{x}$.

$$\zeta(y_{t-1}) = \sum_{\tau=1}^{T_S} \rho_{t\tau} h_\tau \tag{6}$$

$$\rho_{t\tau} = \begin{cases} \frac{1}{K} p(x_\tau, c|s_{t-1}) & x_\tau = y_{t-1} \\ 0 & Otherwise \end{cases} \tag{7}$$

where $K$ is the normalization term. Subsequently, we obtain the probabilities of copying $p_c(y_t|y_{1,...,t-1})$ following the Equation 3, in which the target words are replaced by the words in source text, and the vector of $y_{t-1}$ is replaced by $[\mathbf{e}(y_{t-1}), \zeta(y_{t-1})]^T$, where the $\mathbf{e}(y_{t-1})$ is the embedding of last predicted word.

## 4 Experiment Settings

This section starts with discussing how we design our evaluation experiments, followed by the description of training and testing datasets. Then, we introduce evaluation metrics and baselines.

### 4.1 Training Dataset

There are several publicly-available datasets for evaluating keyphrase generation. The largest one came from Krapivin et al. (2008), which contained 2,304 scientific publications. However, this amount of data is unable to train a robust recurrent neural network model. In fact, there are millions of scientific papers available online, each of which contains the keyphrases assigned by authors. Therefore, we collected a large amount of high-quality scientific metadata in Computer Science domain from various online digital libraries, including ACM Digital Library, ScienceDirect, Wiley, Web of Science, and so on. In total, we obtained 567,830 articles after removing duplicates and overlaps, which is 200 times larger than the one of Krapivin et al. (2008). Note that our model is only trained on 527,830 articles since 40,000 publications are held out for training baselines and for building a test dataset.

### 4.2 Testing Datasets

For evaluating the proposed model more comprehensively, four widely-adopted scientific publication datasets are used. In addition, since these datasets only contain few hundreds or thousands of publications, we contribute a new testing dataset **KP20k** with a much larger number of scientific articles. We take the title and abstract as the source text. Each dataset is described in details in the below text.

- **Inspec** (Hulth, 2003): This dataset provides 2,000 paper abstracts. We adopt the 500 testing papers and the corresponding uncontrolled keyphrases for evaluation, and the remaining 1,500 papers are used for training the supervised baseline models.

- **Krapivin** (Krapivin et al., 2008): This dataset provides 2,304 papers with full-text and author-assigned keyphrases. However, the author did not mention how to split testing data, so we simply select the first 400 papers in alphabetical order as the testing data, and the remaining papers are used to train the supervised baselines.

- **NUS** (Nguyen and Kan, 2007): We use the author-assigned keyphrases and treat all 211 papers as the testing data. Since the NUS dataset did not specifically mention the ways of splitting training and testing data, the results of the supervised baseline models are obtained through a five-fold cross validation.

- **SemEval-2010** (Kim et al., 2010): 288 articles are collected from ACM Digital Library. 100 articles are used for testing and the rest are for training supervised baselines.

- **KP20k**: We build a new testing dataset that contains the title, abstract and keyphrases of 20,000 scientific articles in Computer Science. They are randomly selected from our obtained 567,830 articles. For the supervised baselines, another 20,000 articles are randomly selected for training. Therefore, our proposed model is trained on 527,830 that holds out these 40,000 articles.

### 4.3 Implementation Details

In total, there are 2,780,316 ⟨text, keyphrase⟩ pairs for training, in which text refers to the concatenation of the title and abstract of a publication, and keyphrase indicates an author-assigned keyword. The text pre-processing steps including tokenization, lowercasing and replacing all digits with symbol ⟨digit⟩ are applied. Two encoder-decoder models are trained, one with only attention mechanism (RNN) and one with both attention and copy mechanism enabled (CopyRNN). For both models, we choose the top 50,000 frequently-occurred words as our vocabulary, the dimension of embedding is set to 150, and the dimension of hidden layers is set to 300. Models are optimized using Adam (Kingma and Ba, 2014) with initial learning rate $= 10^{-4}$, gradient clipping $= 0.1$ and dropout rate $= 0.5$. The max depth of beam search is set to 6, and the beam size is set to 200. In the generation of keyphrases, we find that the model tends to assign higher probabilities for shorter keyphrases,

whereas most of keyphrases contain more than two words. To resolve this problem, we apply a simple heuristic by preserving only the first single-word phrase (with the highest generating probability) and removing the rest.

### 4.4 Baseline Models

Four unsupervised algorithms (Tf-Idf, TextRank (Mihalcea and Tarau, 2004), SingleRank (Wan and Xiao, 2008), ExpandRank (Wan and Xiao, 2008)) and two supervised algorithms (KEA (Witten et al., 1999) and Maui (Medelyan et al., 2009a)) are adopted as baselines. We set up the four unsupervised methods following the optimal settings in (Hasan and Ng, 2010) and the two supervised methods following the default setting as specified in their papers.

### 4.5 Evaluation Metric

Three evaluation metrics, the macro-averaged *precision*, *recall* and *F-measure* ($F_1$) are employed for measuring the algorithm performance. Follow the standard definition, precision is defined as the number of correctly-predicted keyphrases over the number of all predicted keyphrases, recall is computed by the number of correctly-predicted keyphrases over the total data records. Note that, when determining the match of two keyphrases, we use Porter Stemmer for pre-processing.

## 5 Results and Analysis

We conduct an empirical study on three different tasks to evaluate our model.

### 5.1 Predicting Present Keyphrases

This is the same as the keyphrase extraction task in prior studies, in which we would like to analyze how well our proposed model perform on the commonly-defined task. To make a fair comparison, we only consider the present keyphrases for evaluation in this task. Table 2 provides the performances of the six baseline models, as well as our proposed models (i.e., RNN and CopyRNN). For each method, the table lists its F-measure at top 5 and top 10 predictions on the five datasets. The best scores are highlighted in bold and the underlines indicate the second best performances.

The results show that the four unsupervised models (Tf-idf, TextTank, SingleRank and ExpandRank) perform robust across different datasets. The ExpandRank fails to return any result on the KP20k dataset due to its high time com-

plexity. The measures on NUS and SemEval here are higher than the ones reported in (Hasan and Ng, 2010) and (Kim et al., 2010), probably because we utilized the paper abstract instead of the full-text for training, which may filter out some noisy information. The performances of the two supervised models (i.e., Maui and KEA) are unstable on some datasets, but Maui achieves the best performances on three datasets among all the baseline models.

As for our proposed keyphrase prediction approaches, the RNN model with the attention mechanism does not perform as well as we expected. It might be because the RNN model only concerns on finding the hidden semantics behind the text, which may tend to generate keyphrases or words that are too general and may not necessarily referring to the source text. In addition, it fails to recall keyphrases that contain out-of-vocabulary words (since the RNN model only takes the top 50,000 words in vocabulary). This indicates that a pure generative model may not fit the extraction task, and we need to further link back to the language usage in the source text. Indeed, the copyRNN model, by considering more contextual information, significantly outperforms not only the RNN model but also all baselines, exceeding the best baselines by more than 20% on average. This result demonstrates the importance of source text for extraction task. Besides, nearly 2% of all the correct predictions contain out-of-vocabulary words.

The example in Figure 1(a) shows the result of predicted present keyphrases by RNN and Copy-RNN for an article about video search. We see that both models can generate phrases that related to the topic of information retrieval and video. However most of RNN predictions are high-level terminologies, which are too general to be selected as keyphrases. The CopyRNN, on the other hand, predicts more detailed phrases like "video metadata" and "integrated ranking". An interesting bad case is, "rich content" is coordinate with a keyphrase "video metadata", and the CopyRNN puts it into prediction mistakenly.

### 5.2 Predicting Absent Keyphrases

As stated, one important motivation for this work is that we are interested in the proposed model's capability for predicting absent keyphrases based on the "understanding" of content. It is worth noting that such prediction is a very challenging task, and, to the best of our knowledge, no ex-

| Method | Inspec | | Krapivin | | NUS | | SemEval | | KP20k | |
|---|---|---|---|---|---|---|---|---|---|---|
| | $F_1@5$ | $F_1@10$ | $F_1@5$ | $F_1@10$ | $F_1@5$ | $F_1@10$ | $F_1@5$ | $F_1@10$ | $F_1@5$ | $F_1@10$ |
| Tf-Idf | 0.221 | 0.313 | 0.129 | 0.160 | 0.136 | 0.184 | 0.128 | 0.194 | 0.102 | 0.126 |
| TextRank | 0.223 | 0.281 | 0.189 | 0.162 | 0.195 | 0.196 | 0.176 | 0.187 | 0.175 | 0.147 |
| SingleRank | 0.214 | 0.306 | 0.189 | 0.162 | 0.140 | 0.173 | 0.135 | 0.176 | 0.096 | 0.119 |
| ExpandRank | 0.210 | 0.304 | 0.081 | 0.126 | 0.132 | 0.164 | 0.139 | 0.170 | N/A | N/A |
| Maui | 0.040 | 0.042 | 0.249 | 0.216 | 0.249 | 0.268 | 0.044 | 0.039 | 0.270 | 0.230 |
| KEA | 0.098 | 0.126 | 0.110 | 0.152 | 0.069 | 0.084 | 0.025 | 0.026 | 0.171 | 0.154 |
| RNN | 0.085 | 0.064 | 0.135 | 0.088 | 0.169 | 0.127 | 0.157 | 0.124 | 0.179 | 0.189 |
| CopyRNN | **0.278** | **0.342** | **0.311** | **0.266** | **0.334** | **0.326** | **0.293** | **0.304** | **0.333** | **0.262** |

Table 2: The performance of predicting present keyphrase of various models on five benchmark datasets

isting methods can handle this task. Therefore, we only provide the RNN and copyRNN performances in the discussion of the results of this task. Here, we evaluate the performance with the recall of top 10 and top 50 results, to see how many absent keyphrases can be correctly predicted. We use the absent keyphrases in the testing datasets for evaluation.

| Dataset | RNN | | CopyRNN | |
|---|---|---|---|---|
| | R@10 | R@50 | R@10 | R@50 |
| **Inspec** | 0.031 | 0.061 | **0.047** | **0.100** |
| **Krapivin** | 0.095 | 0.156 | **0.113** | **0.202** |
| **NUS** | 0.050 | 0.089 | **0.058** | **0.116** |
| **SemEval** | 0.041 | 0.060 | **0.043** | **0.067** |
| **KP20k** | 0.083 | 0.144 | **0.125** | **0.211** |

Table 3: Absent keyphrases prediction performance of RNN and CopyRNN on five datasets

Table 3 present the recalls of the top 10/50 predicted keyphrases for our RNN and Copy-RNN models, in which we observe that the Copy-RNN can, on average, recall around 8% (15%) of keyphrases at top 10 (50) predictions. This indicates that, to some extent, both models can capture the hidden semantics behind the textual content and make reasonable predictions. In addition, with the advantage of features from the source text, the CopyRNN model also outperforms the RNN model in this condition, though not improve as much as the present keyphrase extraction task. An example is shown in Figure 1(b), in which we see that two absent keyphrases "video retrieval" and "video indexing" are correctly recalled by both models. The interesting thing is, the term "indexing" does not appear in the text, but the models

may detect the information "index videos" in the first sentence and paraphrase it to the target phrase. And the CopyRNN successfully predicts another two keyphrases by capturing the detailed information from the text (highlighted text segments).

| Model | $F_1$ | Model | $F_1$ |
|---|---|---|---|
| **Tf-Idf** | 0.270 | **ExpandRank** | 0.269 |
| **TextRank** | 0.097 | **KeyCluster** | 0.140 |
| **SingleRank** | 0.256 | **CopyRNN** | 0.164 |

Table 4: Keyphrase prediction performance of CopyRNN on DUC-2001. The model is trained on scientific publication and evaluated on news.

## 5.3 Transfer to News Articles

The RNN and CopyRNN are supervised models, and they are trained on data in specific domain and writing style. However, with sufficient training on a large-scale dataset, we expect the models to be able to learn universal language features that are effective in other corpus as well. Thus in this task, we will test our model on another type of text, to see whether the model would work when being transferred to a different environment.

We utilize the popular news article dataset **DUC-2001** (Wan and Xiao, 2008) for analysis. The dataset consists of 308 news articles and 2,488 manually annotated keyphrases. The result is shown in Table 4, from which we could see that the CopyRNN could extract a portion of correct keyphrases from a unfamiliar text. Compare to the results reported in (Hasan and Ng, 2010), the performance of CopyRNN is better than TextRank (Mihalcea and Tarau, 2004) and KeyCluster (Liu et al., 2009), but lags behind the other

**Title:** Towards content-based **relevance ranking** for **video search**
**Abstract:** Most existing web **video search** engines index videos by file names, URLs, and surrounding texts. These types of **video metadata** roughly describe the whole video in an abstract level without taking the rich content, such as semantic content descriptions and speech within the video, into consideration. Therefore the relevance ranking of the video search results is not satisfactory as the details of video contents are ignored. In this paper we propose a novel relevance ranking approach for Web-based video search using both **video metadata** and the rich content contained in the videos. To leverage real content into ranking, the videos are segmented into shots, which are smaller and more semantic-meaningful retrievable units, and then more detailed information of video content such as semantic descriptions and speech of each shots are used to improve the retrieval and ranking performance. With **video metadata** and content information of shots, we developed an **integrated ranking** approach, which achieves improved ranking performance. We also introduce machine learning into the ranking system, and compare them with IR-model (information retrieval model) based method. The evaluation results demonstrate the effectiveness of the proposed ranking methods.

**(a) Present Keyphrase**
**RNN:** 1. information retrieval; **2. video search**; 3. search engine; 4. video content; 5. machine learning; 6. web video; 7. content based; 8. semantic content; 9. web based video; 10. web based
**CopyRNN:** 1. information retrieval; **2. video search**; 3. ranking; 4. machine learning; **5. relevance ranking**; **6. video metadata**; **7. integrated ranking**; 8. web video; 9. web video search; 10. rich content

**(b) Absent Keyphrase**
**RNN:** **1. video retrieval**; 2. relevance feedback; 3. video summarization; 4.query expansion; **5.video indexing**; 6.semantic web; 7. multimedia retrieval; 8. image retrieval; 9. web search; 10. query processing
**CopyRNN:** **1. video retrieval**; 2. web search; 3. content ranking; 4. content based retrieval; 5. content retrieval; 6. **video indexing**; 7. relevance feedback; 8. video ranking; 9. semantic web; 10. content based video retrieval; 34. **content based ranking**; 61. **video segmentation**

Figure 1: An example of predicted keyphrase by RNN and CopyRNN. Phrases in bold are correct predictions.

three baselines.

As transferred to corpus in a complete strange type and domain, the model encounters more unknown words and has to rely more on the syntactic features in the text. In this experiment, the CopyRNN recalls 766 keyphrases. 14.3% of them contain out-of-vocabulary words and many names of persons and places are correctly predicted.

# 6 Discussion

Our experimental results demonstrate that the CopyRNN model not only performs well on predicting present keyphrases but also has the ability of generating topical relevant keyphrases that are absent in the text. In a broader sense, this model attempts to map a long text (i.e., paper abstract) with representative short text chunks (i.e., keyphrases), which can potentially be applied to improve information retrieval performance by generating high-quality index terms, as well as assisting user browsing by summarizing long documents into short readable phrases.

So far we have examined our model on scientific publications and news articles, demonstrating that our model has the ability to capture universal language patterns and extract key information from unfamiliar texts. We believe that the models have a greater potential to be generalized to other domains and types, like books, online reviews etc., if it is trained on larger data corpus. Also, we directly apply our model, which is trained on publication dataset, into generating keyphrases for news articles without any adaptive training. We believe that with proper training on news data, the

model would make further improvement.

Additionally, this work mainly studies the problem of discovering core content from textual materials. Here, the encoder-decoder framework is applied to model language; however, such framework can also be extended to locate the core information on other data resources such as to summarize content from images and videos.

# 7 Conclusion and Future Work

In this paper, we propose an RNN-based generative model for predicting keyphrase in scientific text. To the best of our knowledge, this is the first application of the encoder-decoder model to keyphrase prediction task. Our model summarizes phrases based the deep semantic meaning of the text and it is able to handle rarely-occurred phrases by incorporating a copy mechanism. Comprehensive empirical studies demonstrate the effectiveness of our proposed model for generating both present and absent keyphrases for different types of text. Our future work may include the following two directions.

– In this work, we only evaluated the performance of proposed model by conducting offline experiments. In the future, we are interested in comparing the model with human annotators and evaluating the quality of predicted phrases by human judges.

– Our current model does not fully consider the correlation among target keyphrases. It would also be interesting to explore the multiple-output optimization on our model.

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
