# Peer review of "Deep Keyphrase Generation"

_ACL 2017 — decision unknown_

[Official Review · Reviewer 1 · rating 4 · confidence 3]
soundness 3 · originality 4 · clarity 5 · impact 5 · substance 4 · appropriateness 5 · meaningful comparison 5 · presentation format Oral Presentation

This paper proposes to use an encoder-decoder framework for keyphrase
generation. Experimental results show that the proposed model outperforms other
baselines if supervised data is available.

- Strengths:
The paper is well-organized and easy to follow (the intuition of the proposed
method is clear). It includes enough details to replicate experiments. Although
the application of an encoder-decoder (+ copy mechanism) is straightforward,
experimental results are reasonable and support the claim (generation of absent
keyphrases) presented in this paper.

- Weaknesses:
As said above, there is little surprise in the proposed approach. Also, as
described in Section 5.3, the trained model does not transfer well to new
domain (it goes below unsupervised models). One of the contribution of this
paper is to maintain training corpora in good quantity and quality, but it is
not (explicitly) stated.

- General Discussion:
I like to read the paper and would be pleased to see it accepted. I would like
to know how the training corpus (size and variation) affects the performance of
the proposed method. Also, it would be beneficial to see the actual values of
p_g and p_c (along with examples in Figure 1) in the CopyRNN model. From my
experience in running the CopyNet, the copying mechanism sometimes works
unexpectedly (not sure why this happens).

[Official Review · Reviewer 2 · rating 4 · confidence 4]
soundness 3 · originality 4 · clarity 4 · impact 5 · substance 4 · appropriateness 4 · meaningful comparison 5 · presentation format Oral Presentation

This paper divides the keyphrases into two types: (1) Absent key phrases (such
phrases do not match any contiguous subsequences of the source document) and
(2) Present key phrases (such key phrases fully match a part of the text). The
authors used RNN based generative models (discussed as RNN and Copy RNN) for
keyphrase prediction and copy mechanism in RNN to predict the already occurred
phrases. 

Strengths:

1. The formation and extraction of key phrases, which are absent in the current
document is an interesting idea of significant research interests. 

2. The paper is easily understandable.

3. The use of RNN and Copy RNN in the current context is a new idea. As, deep
recurrent neural networks are already used in keyphrase extraction (shows very
good performance also), so, it will be interesting to have a proper motivation
to justify the use of  RNN and Copy RNN over deep recurrent neural networks. 

Weaknesses:

1. Some discussions are required on the convergence of the proposed joint
learning process (for RNN and CopyRNN), so that readers can understand, how the
stable points in probabilistic metric space are obtained? Otherwise, it may be
tough to repeat the results.

2. The evaluation process shows that the current system (which extracts 1.
Present and 2. Absent both kinds of keyphrases) is evaluated against baselines
(which contains only "present" type of keyphrases). Here there is no direct
comparison of the performance of the current system w.r.t. other
state-of-the-arts/benchmark systems on only "present" type of key phrases. It
is important to note that local phrases (keyphrases) are also important for the
document. The experiment does not discuss it explicitly. It will be interesting
to see the impact of the RNN and Copy RNN based model on automatic extraction
of local or "present" type of key phrases.

3. The impact of document size in keyphrase extraction is also an important
point. It is found that the published results of [1], (see reference below)
performs better than (with a sufficiently high difference) the current system
on Inspec (Hulth, 2003) abstracts dataset. 

4. It is reported that current system uses 527,830 documents for training,
while 40,000 publications are held out for training baselines. Why are all
publications not used in training the baselines? Additionally,        The topical
details of the dataset (527,830 scientific documents) used in training RNN and
Copy RNN are also missing. This may affect the chances of repeating results.

5. As the current system captures the semantics through RNN based models. So,
it would be better to compare this system, which also captures semantics. Even,
Ref-[2] can be a strong baseline to compare the performance of the current
system.

Suggestions to improve:

1. As, per the example, given in the Figure-1, it seems that all the "absent"
type of key phrases are actually "Topical phrases". For example: "video
search", "video retrieval", "video indexing" and "relevance ranking", etc.
These all define the domain/sub-domain/topics of the document. So, In this
case, it will be interesting to see the results (or will be helpful in
evaluating "absent type" keyphrases): if we identify all the topical phrases of
the entire corpus by using tf-idf and relate the document to the high-ranked
extracted topical phrases (by using Normalized Google Distance, PMI, etc.). As
similar efforts are already applied in several query expansion techniques (with
the aim to relate the document with the query, if matching terms are absent in
document).

Reference:
1. Liu, Zhiyuan, Peng Li, Yabin Zheng, and Maosong Sun. 2009b. Clustering to
find exemplar terms for keyphrase extraction. In Proceedings of the 2009
Conference on Empirical Methods in Natural Language Processing, pages
257–266.

2. Zhang, Q., Wang, Y., Gong, Y., & Huang, X. (2016). Keyphrase extraction
using deep recurrent neural networks on Twitter. In Proceedings of the 2016
Conference on Empirical Methods in Natural Language Processing (pp. 836-845).

[Official Review · Reviewer 3 · rating 4 · confidence 4]
soundness 3 · originality 4 · clarity 4 · impact 5 · substance 4 · appropriateness 5 · meaningful comparison 5 · presentation format Poster

- Strengths:

Novel model.  I particularly like the ability to generate keyphrases not
present in the source text.

- Weaknesses:

 Needs to be explicit whether all evaluated models are trained and tested on
the same data sets.  Exposition of the copy mechanism not quite
clear/convincing.

- General Discussion:

This paper presents a supervised neural network approach for keyphrase
generation.  The model uses an encoder-decoder architecture that
first encodes input text with a RNN, then uses an attention mechanism to
generate keyphrases from
the hidden states.  There is also a more advanced variant of the
decoder which has an attention mechanism that conditions on the
keyphrase generated in the previous time step.

The model is interesting and novel. And I think the ability to
generate keyphrases not in the source text is particularly
appealing.  My main concern is with the evaluation:  Are all
evaluated models trained with the same amount of data and evaluated
on the same test sets?              It's not very clear.  For example, on the
NUS data set, Section 4.2 line 464 says that the supervised baselines
are evaluated with cross validation.

Other comments:

The paper is mostly clearly written and easy to follow.  However,
some parts are unclear:

- Absent keyphrases vs OOV.  I think there is a need to distinguish
  between the two, and the usage meaning of OOV should be consistent.  The RNN
models
  use the most frequent 50000 words as the vocabulary (Section 3.4
  line 372, Section 5.1 line 568), so I suppose OOV are words not in
  this 50K vocabulary.              In line 568, do you mean OOV or absent
  words/keyphrases?  Speaking of this, I'm wondering how many
  keyphrases fall outside of this 50K?              The use of "unknown words"
  in line 380 is also ambiguous.  I think it's probably clearer to say that
 the RNN models can generate words not present in the source text as long as
they appear
somewhere else in the corpus (and the 50K vocabulary)

- Exposition of the copy mechanism (section 3.4).  This mechanism has a
  more specific locality than the attention model in basic RNN model.
  However, I find the explanation of the intuition misleading.              If I
  understand correctly, the "copy mechanism" is conditioned on the
  source text locations that matches the keyphrase in the previous
  time step y_{t-1}.  So maybe it has a higher tendency to generate n-grams
seen source text (Figure 1).  I buy the argument that the more sophisticated
  attention model probably makes CopyRNN better than the RNN
  overall, but why is the former model particularly better for absent
  keyphrases?  It is as if both models perform equally well on present
keyphrases.

- How are the word embeddings initialized?